# Enhancing Surface Fault Detection Using Machine Learning for 3D Printed Products

**Vaibhav Kadam** [1], **Satish Kumar** [1,*], **Arunkumar Bongale** [1], **Seema Wazarkar** [1], **Pooja Kamat** [1] and **Shruti Patil** [2]

1 Symbiosis Institute of Technology, Symbiosis International (Deemed University), Lavale, Pune 412115, Maharashtra, India; vaibhav.kadam.mtech2019@sitpune.edu.in (V.K.); arunbongale1980@gmail.com (A.B.); seema.wazarkar@sitpune.edu.in (S.W.); pooja.kamat@sitpune.edu.in (P.K.)
2 Symbiosis Centre for Applied Artificial Intelligence, Symbiosis International (Deemed University), Lavale, Pune 412115, Maharashtra, India; shruti.patil@sitpune.edu.in
* Correspondence: satishkumar.vc@gmail.com

**Abstract:** In the era of Industry 4.0, the idea of 3D printed products has gained momentum and is also proving to be beneficial in terms of financial and time efforts. These products are physically built layer-by-layer based on the digital Computer Aided Design (CAD) inputs. Nonetheless, 3D printed products are still subjected to defects due to variation in properties and structure, which leads to deterioration in the quality of printed products. Detection of these errors at each layer level of the product is of prime importance. This paper provides the methodology for layer-wise anomaly detection using an ensemble of machine learning algorithms and pre-trained models. The proposed combination is trained offline and implemented online for fault detection. The current work provides an experimental comparative study of different pre-trained models with machine learning algorithms for monitoring and fault detection in Fused Deposition Modelling (FDM). The results showed that the combination of the Alexnet and SVM algorithm has given the maximum accuracy. The proposed fault detection approach has low experimental and computing costs, which can easily be implemented for real-time fault detection.

**Keywords:** additive manufacturing; fault detection; fused deposition modelling; machine learning; image analysis

## 1. Introduction

Compared to conventional manufacturing, which often involves machining or other techniques to extract surplus material, additive manufacturing (AM) creates components layer by layer [1]. AM uses CAD/CAM software for model generation, then the model is inputted into a 3D printer for slicing and G&M code generation, after which the 3D printer forms a 3D component. There are several types of AM processes that include fused deposition modeling (FDM), stereolithography (SLA), digital light processing (DLP), selective laser sintering (SLS), etc. [2]. The application range of AM is wide; it is used in the field of manufacturing, healthcare, aerospace engineering, fabrication, fashion, etc. [3,4] Because of the low cost of materials and the state of material available, Fused Deposition Modelling (FDM) is the most popular AM method. Despite the diversity of components AM can produce, AM is still susceptible to various defects due to the material properties and structural diversity of printed components. FDM 3D printers are subjected to multiple defects. During printing, due to material property or process failure, the component gets printed with several defects, such as warping, blistering, porosity, cracking, and residual stress.

The research provides an experimental comparative analysis of real-time defect detection. The objectives of this research are:

- To provide a real-time fault detection system for FDM 3D printers.

- To provide a comparative study of model algorithms for fault detection on their computational accuracy results.
- To provide a density-wise classification of printed components.
- To provide ensemble learning results of model algorithm combinations.

The paper overview includes the system methodology in which the experimental approach, algorithms, and pre-trained model used are explained. Further, it includes the experimental setup used, including the data collection technique and the result obtained by experimental analysis and comparative results of model algorithms.

Different defects that occur in printed components and their causes and effects are shown in Table 1. Out of all the defects in 3D printed components, most are because of the material property or printing technique used. Warping gets introduced during the printing of a long component. The material extrusion layer-by-layer technique used in FDM components often requires post-processing since the printed component has a poor surface finish. Sometimes formation of small voids can lead to crack generation and, later on, failure of a component. Various defects that occur, such as clogging of the nozzle, improper bed leveling, misalignment of the printing platform, lack or loss of adhesion of the print platform etc., can be solved by manual adjustment. These errors are mainly due to carelessness of the operator and are not part of the research.

**Table 1.** Defect analysis.

| Sr No. | Defect | Cause | Effect |
|---|---|---|---|
| 1 | Warping [5] | Improper cooling of the printed component or due to materials used in the process | The printed part swells upward, resulting in a change in the shape of the component. |
| 2 | Blistering | Due to improper cooling of lower layers. | A lower layer of printed component swells outward due to the weights of upper layers. |
| 3 | Porosity [6] | Improper printing process or material used in the process | Very small air bubbles or cavities get a form in the printed component during the printing process |
| 4 | Cracking [6] | Due to small cavities leading to the formation of cracks, stress formation, or uneven heating or cooling of a particular area | Cracks are formed in the component, which can result in the failure of a printed part. |
| 5 | Residual stresses [7] | Stress is induced due to rapid heating or cooling of the material, leading to expansion or contraction. | When residual stress exceeds the limit of tensile strength, then it can lead to the formation of cracks or defects such as warpage |
| 6 | Poor surface finish [8] | Printing technique and material used in the process | Parts produced often require post-processing |
| 7 | Stringing [9] | Due to material property and printing properties. | Parts produced often have strings of extra material attached to them. |
| 8 | Material shrinkage [10] | Materials used in 3D printing have a certain degree of shrinkage. | If material shrinkage is too large, residual stress may occur, which can cause deformation of the part or crack generation. |

## 2. Literature Review

Given that 3D printing is still a relatively new technology in the manufacturing sector, there is limited literature addressing quality issues with 3D printing. H. Gunaydin et al. have stated the different errors that occurs, such as clogging of the nozzle, adhesion problem, vibration or shocks, misalignment of the print platform, etc., which causes loss of material, time, and money [8]. D. Geng and J. Zhao stated the severity of warpage

problems, which are caused due to improper cooling [5]. Apart from machine errors, printing components are subjected to errors due to structural and material shortcomings such as porosity, cracking, residual stresses, etc. [7]. L. Yuan emphasizes the solidification defects in printed components that affect the overall strength of the printed part [6].

During the manufacturing process, the ongoing process or system component often interacts with the environment, humans, and various parameters that affect the physical element. An important factor for monitoring a system is the availability of built-in sensors, but current FDM printers lack these built-in sensors. Therefore, it is a very difficult task for sensing the real-time system state, which is vital for fault detection. Many studies are performed for anomaly detection using a sensor-based approach such as Kousiatza and Karalekas, who used temperature sensors and thermocouples to generate temperature profiles for fault detection [11]. Li et al. provided a sensor-based model for surface anomaly detection [12]. Many works are done using a sensor-based model for fault detection. However, almost all FDM machines currently have few sensing capabilities that are either inaccessible to users or are not equipped with feedback measurement systems for process correction [13]. For diagnosing a single defect, sensor-based monitoring systems need several sensors. In contrast, only a few sensors can precisely track and recognize product quality during the actual process. Finding the sensor's perfect position is difficult since data gathering accuracy depends on the sensors' position.

The majority of the defects are detectable by the naked eye. Still, it is difficult to consistently monitor the process by sight, making it difficult to detect errors on time. Some errors go unnoticed in the sensor-based approach as it does not consider the errors that occur in layers while printing. This study focuses on anomaly detection using a camera using the layer-wise approach. Computational Image Analysis is an interdisciplinary area that allows computers to interpret images and video frames at a higher level. Computer vision (CV) is typically a difficult task since it focuses on various issues such as image segmentation, object tracking in a video stream, feature extraction, and motion tracking [14]. Many printers now include a monitoring camera that can be streamed to a website or a smartphone app; this makes it much easier to keep a closer eye on the printer and ensure that nothing is wrong. However, human interaction is still required, and the additive manufacturing process is not as automated as possible. To avoid human interaction or minimize it as much as possible, machine learning (ML) can play an important role since ML provides various algorithms for classification, segmentation, error detection, etc. [15]. Many works are done in the area of anomaly detection using ML. Machine learning [16] can play a critical role in developing multi-level predictive models for the AM process. Many machine learning models have been investigated for specific processes and applications to find faults in the AM process [17,18]. N. Silaparasetty stated the overview of ML, deep learning, and big data [19]. A. Dey explained all the techniques of ML and different algorithms with structure, ML provides wide range of algorithm that can be used for fault detection [20]. Zhang et al. implemented ML model and computational data to control powder quality in metal AM processes obtained from the Discrete Element Method [21]. Stoyanov et al. used an ML model to improve the electronics component generated by 3D inkjet printing [22]. Many works use ML architectures combined with acoustics or visual monitoring for automatic defect detection during the printing process. Konstantinos Paraskevoudis et al. used a computer vision approach for stringing type error but did not consider layer-wise fault detection [9].

Literature findings are shown in Table 2 in the form of the 3D printer technique, the approach used for study, selection of the model, and accuracy obtained by selected model. Table 2 shows the literature findings in a simplified format; it shows which AM technique is considered for the experimental purpose, such as FDM, SLS, etc. Table 2 visualizes the approach used in research; it may be sensor-based, computer vision-based (with the help of a camera), or any other monitoring or fault detection method. The table also incorporates which technique is used for fault detection, whether it is ML, deep learning,

convolutional neural network (CNN), artificial neural network (ANN), or a combination of these techniques along with their accuracy obtained.

**Table 2.** Literature analysis.

| Sr No. | Author | AM Techniques Considered | Approach | Test Parameter Considered | Model Selection | Work Done | Accuracy |
|---|---|---|---|---|---|---|---|
| 1 | Ugandhar Delli et al. [23] | FDM | Computer vision based (used camera for capturing images) | Standard | ML (SVM) | The study provides CV based approach for anomaly detection. Images for processing are taken at the interval but not continuous. | Not mentioned |
| 2 | Danielle Jaye S. Agron ET AL. [24] | SLS | Sensor-based | Standard | ANN | The study provides a model to monitor and control oxygen levels in SLA. | 96% |
| 3 | Hermann Baumgartl et al. [25] | L-PBF | CV Based (used thermographic camera) | Standard | CNN | This study explores various anomaly detection techniques used in Laser powder-based fusion AM; for detection, different methods are proposed such as melt pool monitoring or off-axis infrared monitoring. This research tries to provide a model for fault detection for low cost using heat maps and infrared imaging. | 96.80% |
| 4 | Luke Scime et al. [26] | L-PBF | CV Based (used camera for capturing images) | Standard | ML | The study provides a computer vision-based approach for detecting anomalies in the powder spreading stage in laser powder-based fusion AM. | Not Mentioned |
| 5 | Zeqing Jin et al. [27] | FDM | CV Based (used camera for capturing images) | Nozzle height- High+, High, Good, and 'Low' | CNN | This paper provides a self-monitoring system for inter-layer imperfections such as warping and delamination defects using the deep learning model. The paper also provides a technique for auto-calibration and pre-diagnosis of defects. | 97.80% (validation) 91% (testing) |
| 6 | S. A. Langeland [14] | FDM | CV Based (used camera for capturing images) | Infill pattern, density | ML algorithm and CNN | The study provides automatic monitoring and anomaly detection system using computer vision. | Not Mentioned |
| 7 | Yaser Banadaki et al. [13] | FDM | CV Based (used camera for capturing images) | Printing speed, temperature | CNN | The study provides a CNN model for the plastic AM process for fault detection. Paper proposed an automated quality grading system for the printed component. | 94% |
| 8 | Konstantinos Paraskevoudis et al. [9] | FDM | CV Based (used camera for capturing images) | Temperature, speed. Layer thickness | CNN | Used computer vision approach for stringing type error. The study provides a deep learning model for predicting and detecting stringing error in a printed component. | 92.70% |
| 9 | Zhixiong Li et al. [12] | FDM | Sensor Based | Feed rate, layer thickness, temperature | ML algorithm | The literature provided a sensor-based model for surface anomaly detection in AM. The study includes an ensemble model for predicting surface roughness. | 55–59% |
| 10 | K. Wasmer et al. [28] | PBF | Sensor Based | Scanning velocity | Machine Learning and reinforcement learning | The research tried to monitor AM components using acoustic emission and reinforced learning for mass production of AM components with the same standards. | 74–82% |
| 11 | Ikenna A. Okaro et al. [29] | L-PBF | Sensor Based | Standard | ML Semi-supervised approach | A semi-supervised approach is used | 77% |

Many studies include monitoring based on sensors, which involves finding the perfect location for the sensor since sensor location is an important factor that affects the model's overall efficiency. Very few studies used a layer-wise image capturing approach for fault detection. This study provides a method to identify defects by capturing the layer-wise

photo of the printing process with the help of ML and computer vision. This paper suggests a computer vision system based on machine learning for monitoring the quality characteristics of additive manufacturing processes. The material may overfill or underfill due to the residual pressure of the melted filament inside the extrusion chamber, resulting in visible surface defects or unseen internal defects, leading to degradation of the product's quality. System failures can be predicted and flagged by the monitoring system in the earliest stage of the proposed monitoring system. For the implementation of the monitoring system, we first trained the model in offline mode by gathering an image dataset, and then tested the predictive model for online AM process monitoring. This capability will transform the 3D printer into a self-inspecting machine capable of inspecting parts as they are being constructed, adding another layer of quality control to the process.

## 3. Methodology

All the work is performed on the FDM-based 3D printer Dreamer; no hardware changes are done except camera mounting for image capturing. Red-Green-Blue (RGB) images are automatically captured with the raspberry-pi camera. All the programming, training, and testing are done in Matlab.

### 3.1. System Methodology

A layer-wise approach is used most of the time, as the defects involved in printed components go unnoticed in the sensor-wise approach for fault detection. Current FDM printers lack inbuilt sensors, which is why this study focuses on layer-wise monitoring of printing components for fault detection. Layer-wise monitoring involves capturing layer-wise images of printed components for training, processing, and classification.

The experimental process flowchart is shown in Figure 1; the study starts with setup formation to capture layer-wise images. In the second stage, an image dataset is created by capturing multiple layer-wise images of the printing component. The prepared image dataset is processed for noise reduction, segmentation, and cropping in the next step. After successfully creating the dataset, a combination of a model algorithm is selected for training and testing. Upon identifying optimal combination, a model is implemented for real-time fault detection.

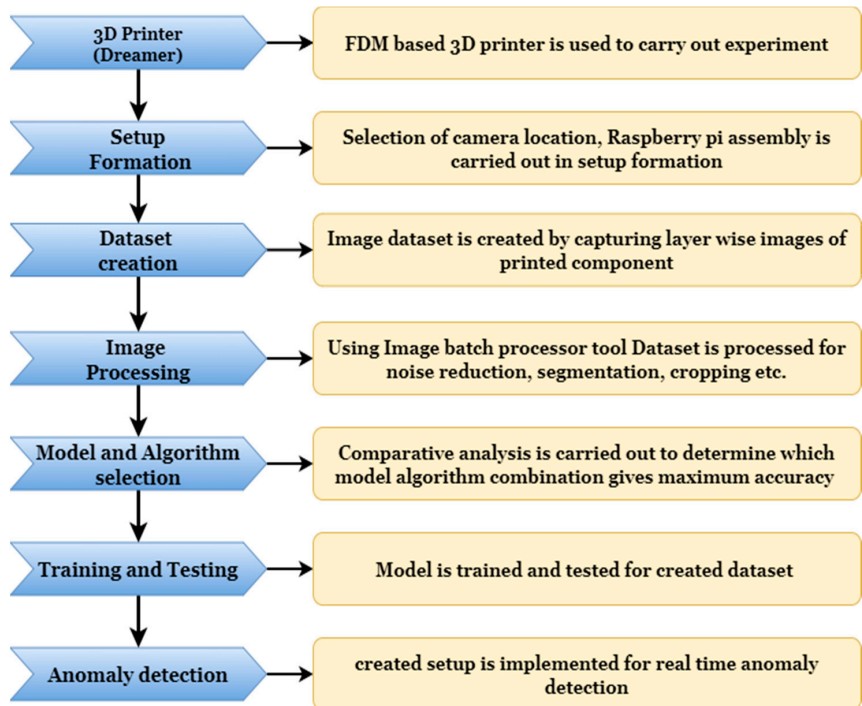

**Figure 1.** Process flowchart.

Pre-trained models are used only for feature extraction purposes. Training and validation are carried out with different algorithms such as Support Vector Machine (SVM), K-Nearest Neighbor (KNN), Random Forest, etc.

Polylactic Acid (PLA) material is selected as the printing material; PLA is a polymer made from corn starch and other organic materials. PLA becomes slightly more liquid and harder during printing than ABS. As a consequence, the prints are typically more informative than Acrylonitrile Butadiene Styrene (ABS) prints. PLA and ABS are almost indistinguishable visually, with PLA being slightly shinier.

### 3.2. Algorithm Study

Various current state-of-the-art algorithms are available, with different training speeds, accuracy, and testing speeds in benchmark datasets. Since the aim of the model is to be deployed in a live setting, we considered the need to strike a balance between good accuracies and quick detection in our case.

MATLAB is used for feature extraction and anomaly detection since it provides a variety of pre-trained models, image processing tools, algorithms readily available with a handful of command lines. A pre-trained model is used for training; Alexnet, Googlenet, Resnet18, Resnet50, and Efficientnet-b0 are the different pre-trained models used for feature extraction and training purposes. The image dataset is pre-processed as per the model's requirement since different pre-trained models have different image input sizes.

Pre-trained models are only used for feature extraction purposes. For training and classification, different algorithms are used to improve model accuracy further. The different algorithms used are

1.  K-Nearest Neighbor (KNN): In KNN, the labeled dataset created for training purposes is fed into the classifier/learner, then the learner classifies the sets of data inputted. K most correlated data from the training set is chosen. Most of K is selected, and test data is assigned to a new class [30]. Figure 2 shows the architecture of the K-nearest neighbor classifier [31].
2.  Support Vector Machine (SVM): SVM is another state of art algorithm which is mostly used for categorization. SVM is based on the concept of calculating margins. It is used to separate groups of data by drawing a line in between. The margins are selected such that there is a minimum difference between margin and labeled classes resulting in reducing classification error [32]. Figure 3 shows the architecture of the support vector machine classifier [33].
3.  Naive Bayes: Naive Bayes is primarily employed for clustering and classification. The Bayesian network is mainly used for probability distribution, which is described by direct acyclic graphs (DACG). Nodes in the Bayesian network represent the variable, and the connecting arc means probabilistic dependency between variables. The conditional probability is used in the underlying architecture of Naive Bayes. It produces trees dependent on the likelihood of them occurring. Bayesian Network is another name for these trees [34]. Figure 4 shows the Naive Bayes classifier structure [35].
4.  Decision Tree: A decision tree is made of nodes and branches; it is primary used for classification purposes. It sorts the attribute as per their values and groups them together. A node represents an attribute that needs to be categorized, and a branch represents a value taken by a node [36]. Figure 5 shows the basic architecture of the decision tree [37].
5.  Random Forest: As per the name, a random forest is made of many decision trees employed together for working, resulting in an ensemble. Each tree in a random forest generates class data prediction, depending upon the majority of votes forecast for the model [38]. Figure 6 shows the basic architecture of random forest [39].

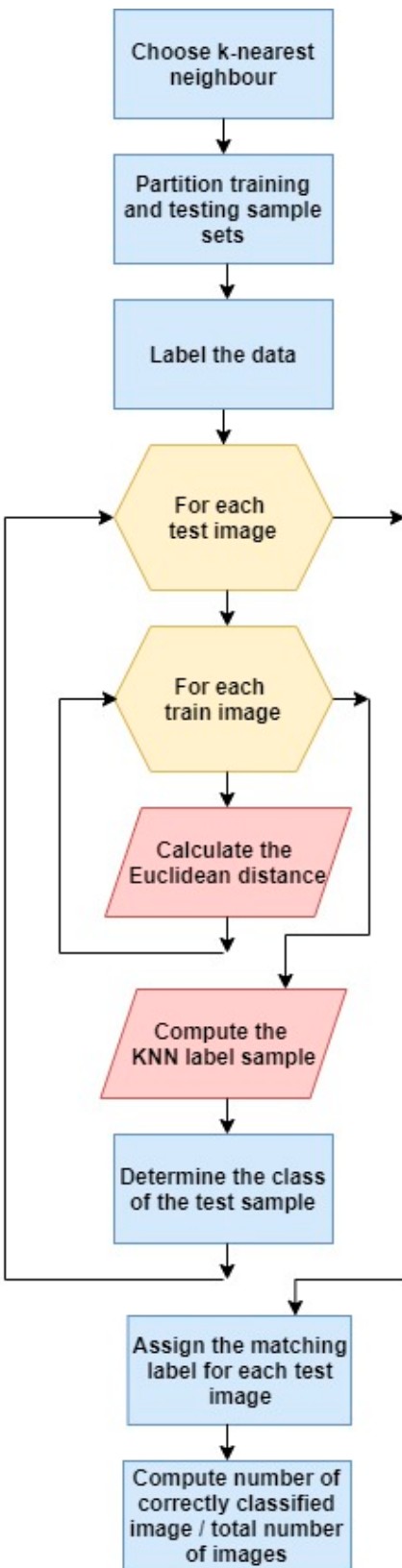

**Figure 2.** Flowchart of KNN algorithm.

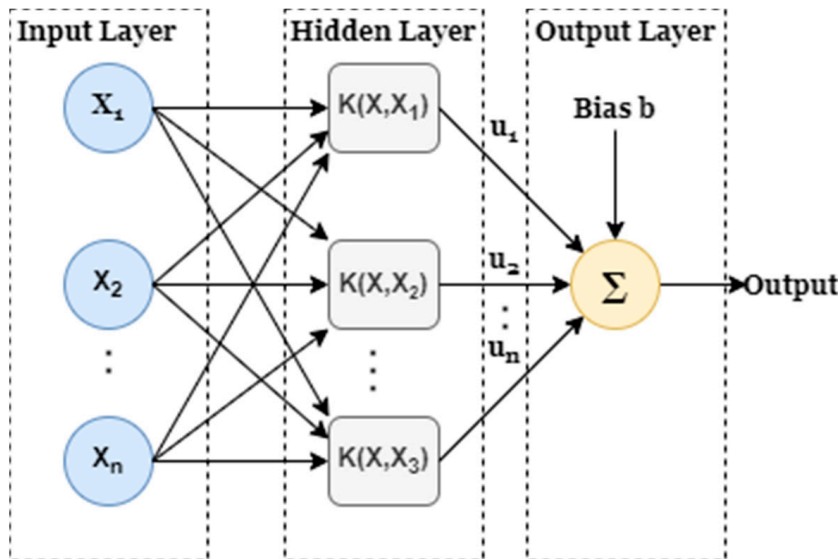

**Figure 3.** Architecture of SVM algorithm.

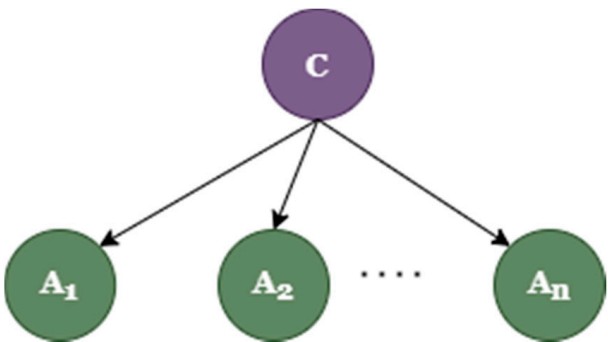

**Figure 4.** Naive Bayes classifier structure.

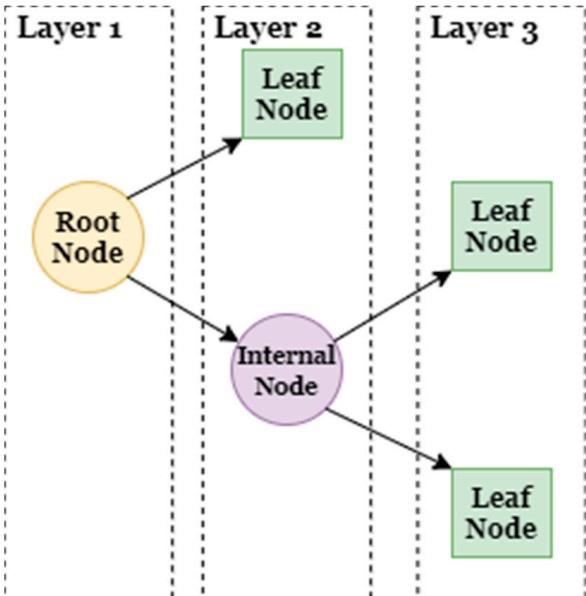

**Figure 5.** Architecture of the Decision Tree algorithm.

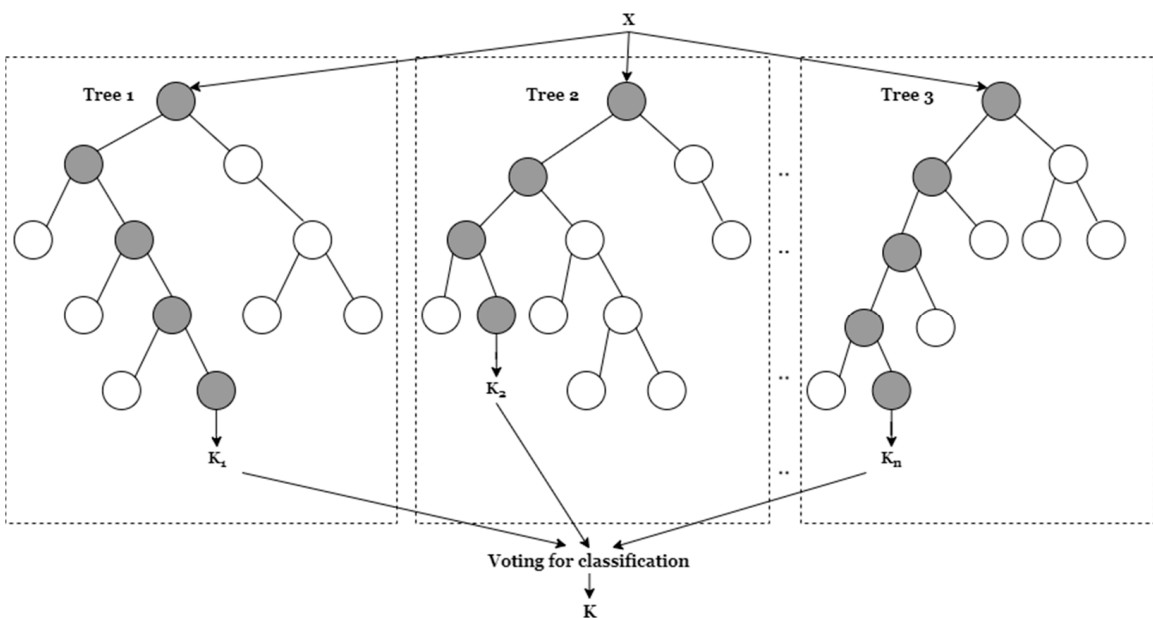

**Figure 6.** Architecture of the random forest algorithm.

A further comparative study is performed to identify which model gives maximum accuracy. As mentioned above, different pre-trained models are used in combination with a different algorithm to calculated combined accuracy. By this comparative study, we are able to find out the quickest and most accurate model and algorithm combination for our dataset.

*3.3. Ensemble Learning*

The art of integrating a diverse set of learners (individual models, algorithms) to boost the model's stability and predictive capacity is known as ensemble learning. It is a powerful tool for improving model efficiency and accuracy. A pre-trained model is an ensemble with different algorithms used to increase model accuracy. In ensemble learning, predictive results obtained by different classification algorithms are compared for count generation. If the count value of the non-defective result (Good) is greater than two out of five, then the ensemble result is selected as non-defective, or else, it is chosen as defective (Bad). Figure 7 shows the process flowchart of ensemble learning.

*3.4. Evaluation Principle*

For evaluation of defective and non-defective layers, images are captured of both defective and non-defective layers. After successfully creating a dataset, a dataset is labeled as good and bad. Good for non-defective layer and bad for defective layer. Labeling is done manually as per eye inspection. Errors observed during labeling the dataset are improper filling of material, improper pattern development, and stringing problem. Two different datasets are created for defective and non-defective components, and error detection is carried out for errors observed and occurred. Figure 8 shows the example of non-defective layers or defect-free layers in the printing process. As shown in Figure 8, layer-wise images of each layer are captured and labeled accordingly.

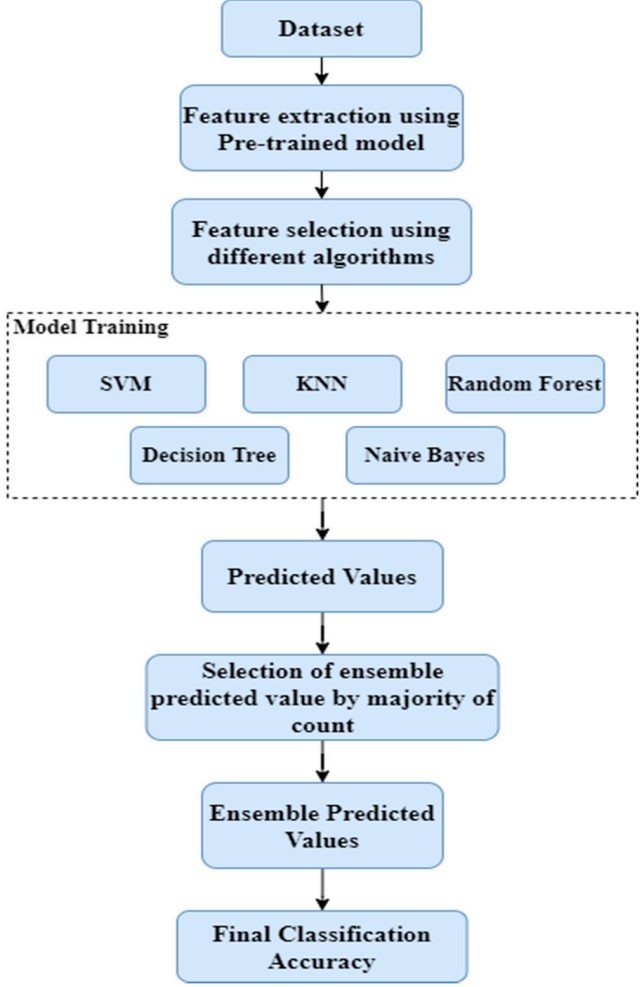

**Figure 7.** Ensemble learning approach.

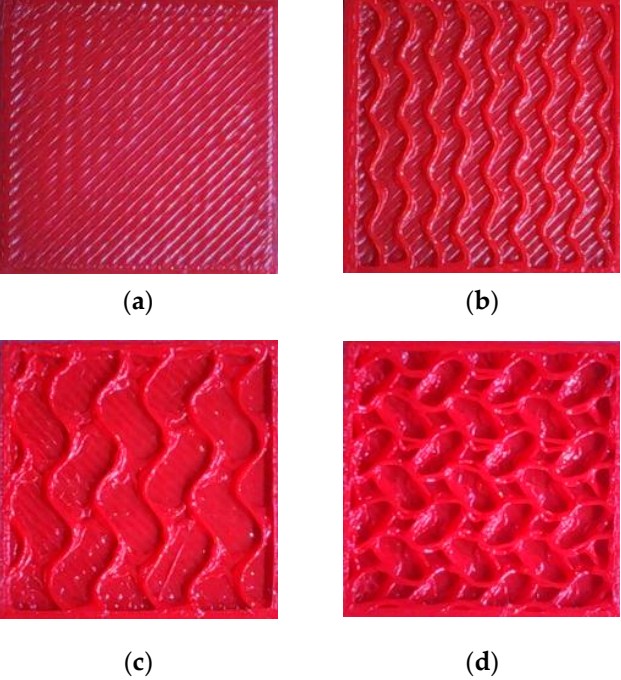

**Figure 8.** Visualization of non-defective layers.

Visualization of the defective layer or defect that occurred in the printing layer is shown in Figure 9. Most defects have occurred in the first and last two to three layers in the component; much less errors are observed in the pattern filling part.

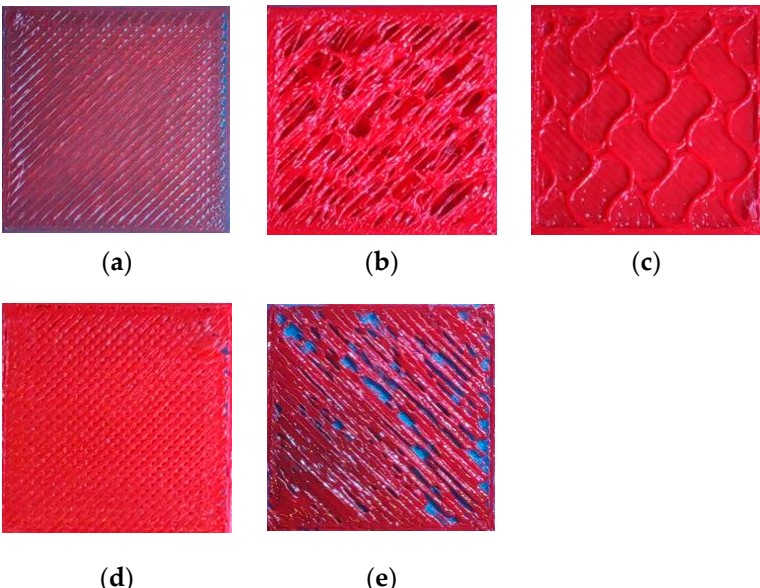

Figure 9. Visualization of defective layers.

*3.5. Density Wise Classification*

The study also includes the identification of components based on process parameters. Parameter variation consists of temperature, printing speed, and density but a part to be printed or layers to be printed is the same for temperature and speed variation. Due to this reason, one cannot identify printing components based on these parameters. However, density-wise identification of printed parts is possible. For density-wise classification, again, these pre-trained models are used, but for this time, pre-trained models are used for feature extraction and also for training and testing purposes.

*3.6. Pre-Trained Models Used*

1. Alexnet: Alexnet is an eight-layer convolutional neural network (CNN) in which the first five layers are convolutional, and the last three are fully connected layers [40]. It can classify 1000 different classes; the input image size for Alexnet is 227 by 227. Figure 10 shows the architecture of the pre-trained model Alexnet.
2. Googlenet: Googlenet is a 22-layer convolutional neural network, the image input size for this network is 224 by 224 [41]. It can predict classes up to 1000 classes. Figure 11 shows a simplified block diagram of the Googlenet architecture.
3. Resnet18: Resnet is a short form for the residual net; it is a classic neural network, and as the name suggests, it is an 18-layer network [42]. It takes image input size as 224 by 224. It takes an image in the form of Red-Green-Blue (RGB). Figure 12 shows the architecture of Resnet18.
4. Resnet50: As the name suggests, it is a 50-layer deep CNN [43]; the required image input size for this network is also 224 by 224. It has a 1-maxpool layer, 1-average pool layer, and 48 convolutional layers. Figure 13 shows the basic architecture of Resnet50.
5. Efficientnet-b0: there are 237 layers in Efficientnet [44]; it can train a database of up to 1000 classes. The image input size for this network is 224 by 224, and the required format is RGB. Figure 14 shows the basic architecture of Efficientnet-b0.

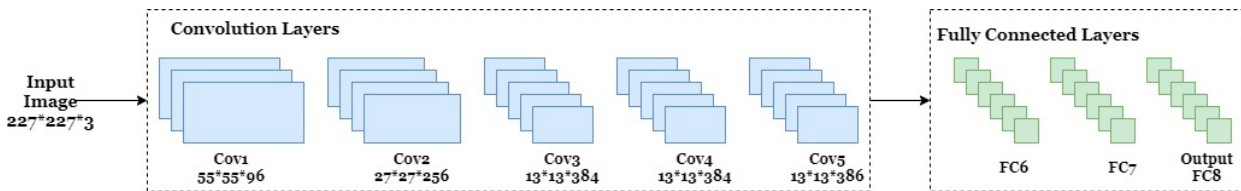

**Figure 10.** Architecture network of Alexnet.

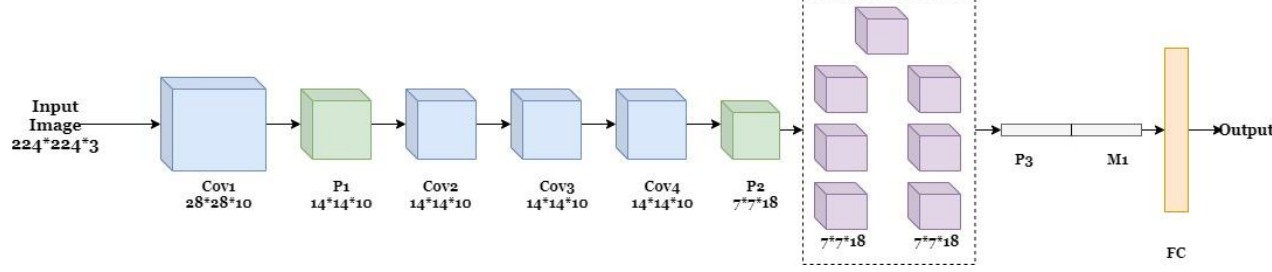

**Figure 11.** Architecture of Googlenet.

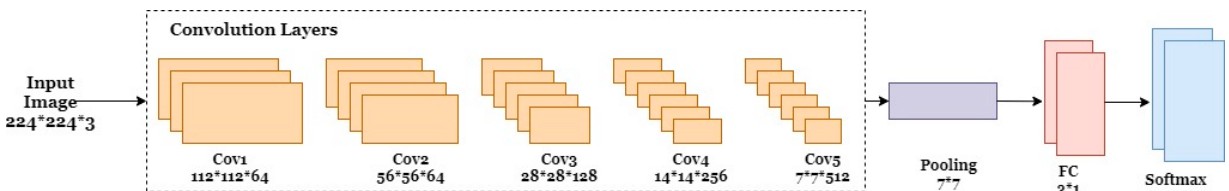

**Figure 12.** Architecture of Resnet18.

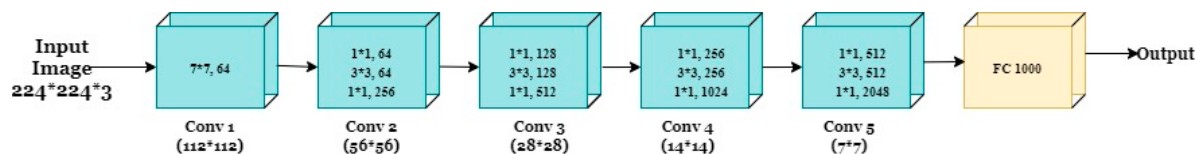

**Figure 13.** Architecture of Resnet50.

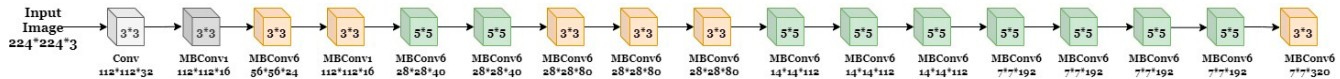

**Figure 14.** Architecture of Efficientnet-b0.

## 4. Materials and Method

### 4.1. Experimental Setup

The experimental setup is shown in Figure 15; for this study, the FDM-based 3D printer (Dreamer) is used. An 8MP Raspberry pi camera captures images mounted below the nozzle head beside the nozzle. The camera's position is a very important factor since everything depends upon the quality of the images captured. Camera position is decided after placing it in different locations and capturing images, and images are compared for quality, then optimal location is finalized. Raspberry pi 4B is used for the processing and is connected to the Raspberry pi 7-inch display; the Pi-Camera is connected to the raspberry-pi via a flex cable.

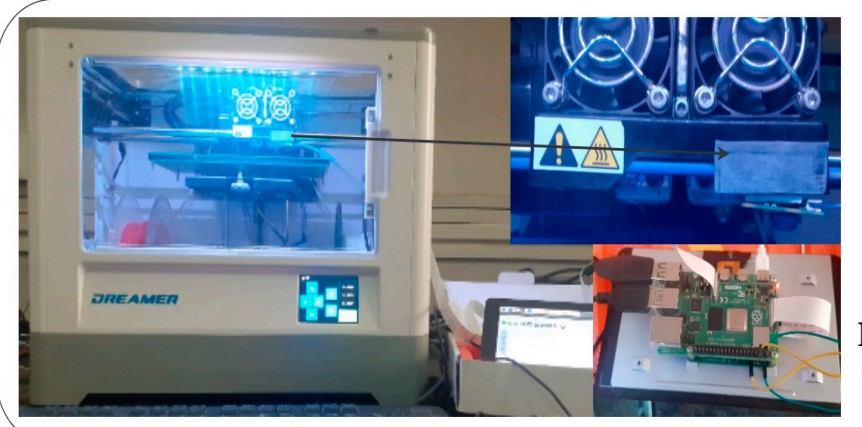

**Figure 15.** Experimental setup.

*4.2. Data Collection and Annotation*

A cube with (25 mm × 25 mm × 5 mm) size is selected as the test object. Layer-wise images are taken with 4 to 5 images of each layer. A python program is created to capture images, which captures an image by pressing any key or with a mouse click. Automatic capturing of images between the finite interval of time is avoided since the printing time for base layers varies compared to pattern filling. The machine takes a long time to fill the first and last two layers of a component compared to the middle layers. A total of 1700 image datasets are created with both defective and non-defective images by varying process parameters.

After creating the dataset, images are processed for noise reduction, segmentation, and size optimization. A raw image cannot be fed to the model as it affects model accuracy and performance. Figure 16 shows the raw image and the image after processing. Images are cropped to reduce noise in images. For processing images, an image batch processing tool is used, which is available in Matlab.

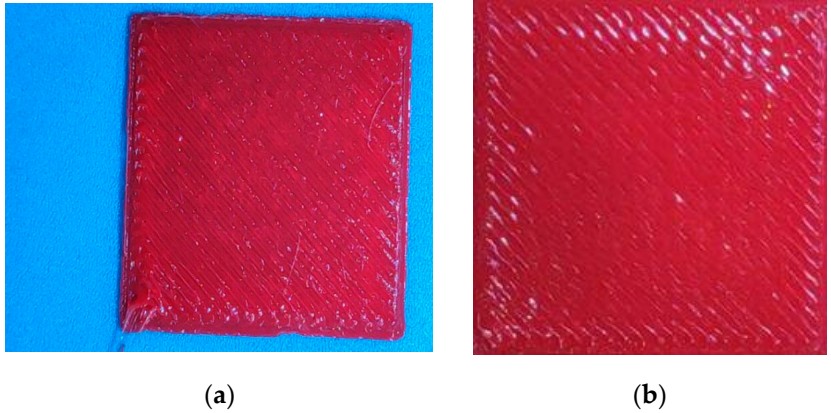

(**a**)          (**b**)

**Figure 16.** Raw image (**a**) vs. processed image (**b**).

Varying process parameters, different images are obtained. Sample objects are created by parameter variations such as temperature, printing speed, and density. A total of 32 variants are created by varying these parameters. A pattern used is 3D infill, which is the same for all printed components. Table 3 shows the printing object with test parameter variation.

**Table 3.** Test parameters.

| Sr No. | Density (%) | Temperature | Printing Speed |
|---|---|---|---|
| 1 | 15 | 200, 210, 220 | 60, 70, 80 |
| 2 | 20 | 200, 210, 220 | 60, 70, 80 |
| 3 | 25 | 200, 210, 220 | 60, 70, 80 |
| 4 | 30 | 200, 210, 220 | 60, 70, 80 |
| 5 | 35 | 200, 210, 220 | 60, 70, 80 |
| 6 | 40 | 200, 210, 220 | 60, 70, 80 |
| 7 | 45 | 200, 210, 220 | 60, 70, 80 |
| 8 | 50 | 200, 210, 220 | 60, 70, 80 |

## 5. Results

The comparative results of different pre-trained models and algorithms are obtained for real-time monitoring and fault detection. Accuracy and loss are obtained for model algorithms; the formulae used are shown in Table 4.

**Table 4.** Formulae used for calculations.

| Accuracy | $\frac{\sum \text{ True Positive} + \sum \text{ True Negative}}{\sum \text{Total Population}}$ |
|---|---|
| Error | 100-Accuracy |

### 5.1. Model Accuracy

Comparative analysis is carried out to find out which pre-trained model algorithm combination gives maximum accuracy. Every pre-trained model algorithm combination is implemented for our dataset, and accuracy results are obtained.

The combined accuracy of the different pre-trained models with various algorithms for our captured dataset is shown in Table 5. From the comparison, it is clear that the Alexnet and Efficientnet-B0 model combined with SVM gives maximum accuracy, which is 99.70%, followed by Resnet50 in combination with SVM, which provides an accuracy of 99.40% with 0.60% loss. From the below comparison, we can observe that SVM gives maximum accuracy with different pre-trained models out of all other algorithms.

**Table 5.** Comparative analysis of different model-algorithm accuracy.

| Sr No. | Algorithm | Alexnet | | Googlenet | | Resnet18 | | Resnet50 | | Efficientnet B0 | |
|---|---|---|---|---|---|---|---|---|---|---|---|
| | | Accuracy (%) | Loss (%) | Accuracy (%) | Loss (%) | Accuracy (%) | Loss (%) | Accuracy (%) | Loss (%) | Accuracy (%) | Loss (%) |
| 1 | SVM | 99.70 | 0.30 | 99.10 | 0.90 | 97.20 | 2.80 | 99.40 | 0.60 | 99.70 | 0.30 |
| 2 | KNN | 99.40 | 0.60 | 98.80 | 1.20 | 98.50 | 1.50 | 99.10 | 0.90 | 99.70 | 0.30 |
| 3 | Random Forest | 97.20 | 2.80 | 99.10 | 0.90 | 95.40 | 4.60 | 98.50 | 1.50 | 98.80 | 1.20 |
| 4 | Decision Tree | 96.60 | 3.40 | 98.20 | 1.80 | 96.30 | 3.70 | 96.90 | 3.10 | 96.90 | 3.10 |
| 5 | Naive Bayes | 85.90 | 14.10 | 90.00 | 10.00 | 91.10 | 8.90 | 88.00 | 12.00 | 90.20 | 9.80 |

The confusion matrix of Alexnet with SVM is shown in Figure 17a–c, showing the confusion matrix of efficientnet-B0 with SVM and KNN. These combinations are giving maximum accuracy for our dataset.

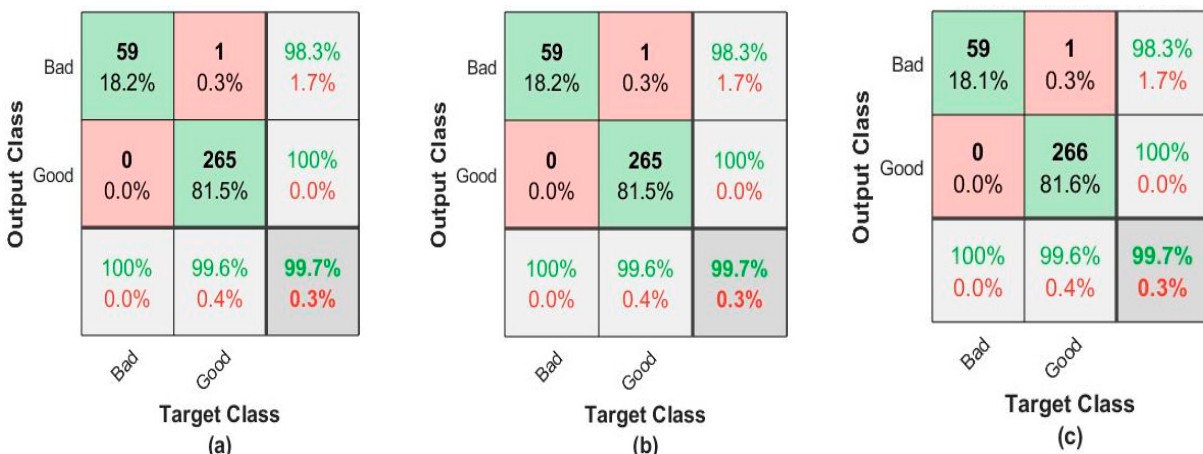

**Figure 17.** Confusion matrix giving the highest accuracy: (**a**) Confusion matrix of Alexnet with SVM, (**b**) Confusion matrix of Efficientnet-B0 with KNN, (**c**) Confusion matrix of Efficientnet-B0 with SVM.

The graphical visualization of comparative accuracy of the model algorithm study is shown in Figure 18. Graphic visualization makes it easy to identify maximum accuracy concerning algorithm and different pre-trained models.

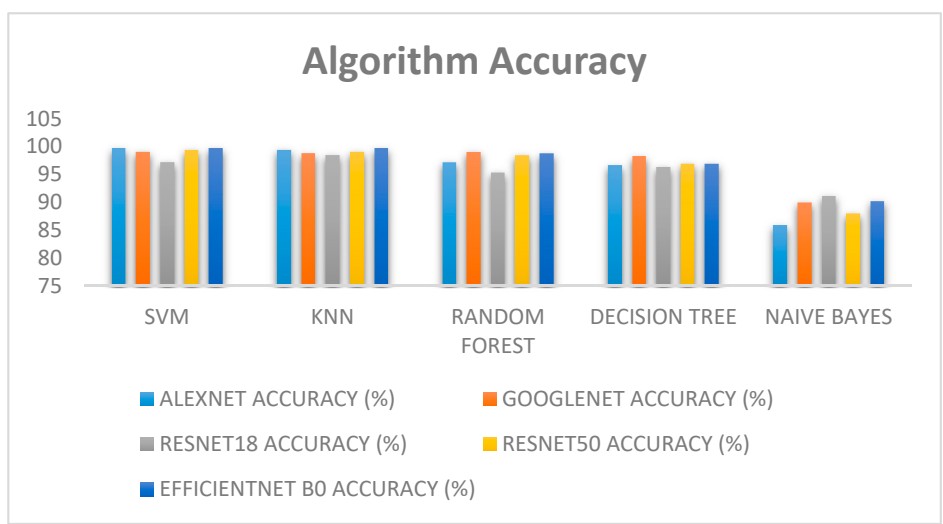

**Figure 18.** Comparative analysis of different model algorithm accuracy.

### 5.2. Ensemble Learning

The ensemble accuracy of different pre-trained models is shown in Table 6; we can observe that Alexnet gives 100% accuracy, followed by resnet18 with 99.40% with a 0.60% loss. Googlenet provides the least accuracy by ensemble learning with a different algorithm, which is 97.80%.

**Table 6.** Accuracy analysis by ensemble learning.

| Sr No. | Model | Accuracy | Loss |
|--------|-------|----------|------|
| 1 | ALEXNET | 100.00% | 0.00% |
| 2 | GOOGLENET | 97.80% | 2.20% |
| 3 | RESNET18 | 99.40% | 0.60% |
| 4 | RESNET50 | 98.80% | 1.20% |
| 5 | EFFICIENTNET-B0 | 99.10% | 0.90% |

Confusion matrix: The confusion matrix obtained by ensemble learning is shown in Figure 19. Figure shows input and output class, namely good and bad. Defective layer im-

age class is named as bad and non-defective layer image class is named as good. Accuracy shown by the confusion matrix is ensemble accuracy obtained by different algorithms.

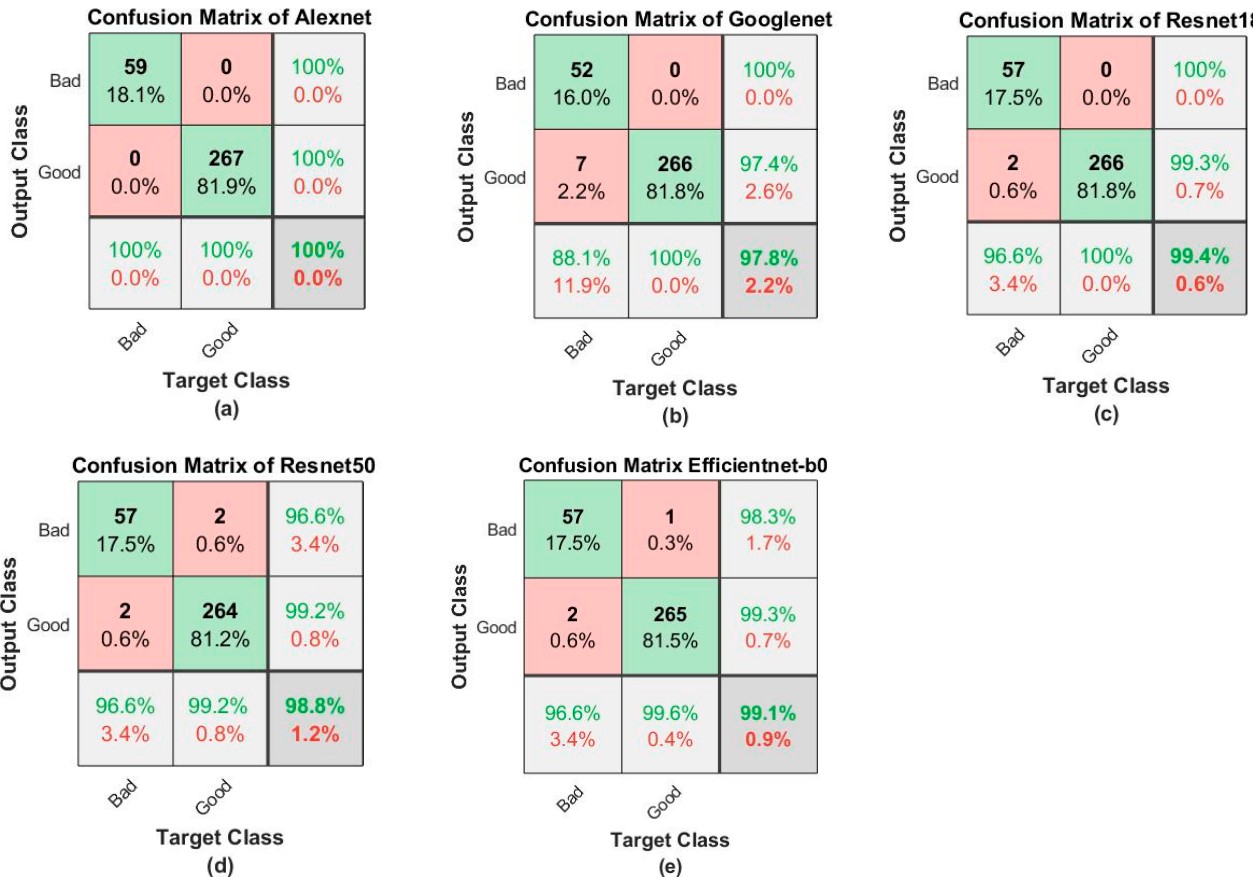

**Figure 19.** Visualization of the ensemble confusion matrix: (**a**) Ensemble confusion matrix of alexnet; (**b**) Ensemble confusion matrix of googlenet; (**c**) Ensemble confusion matrix of resnet18; (**d**) Ensemble confusion matrix of resnet50; (**e**) Ensemble confusion matrix of efficientnet-b0.

### 5.3. Density Wise Classification

Further parameter-wise classification is performed, which includes density-wise classification. Table 7 shows the accuracy of pre-trained models used for density-wise classification. We can observe that resnet50 gives 100% accuracy, which is outstanding, followed by Alexnet and Efficientnet, delivering 99.22% with 0.78% loss.

**Table 7.** Models accuracies of density wise classification.

| Sr No. | Model | Accuracy | Loss |
|--------|-------|----------|------|
| 1 | ALEXNET | 99.22% | 0.78% |
| 2 | GOOGLENET | 97.66% | 2.34% |
| 3 | RESNET18 | 97.66% | 2.34% |
| 4 | RESNET50 | 100% | 0% |
| 5 | EFFICIENTNETB0 | 99.22% | 0.78% |

Images are inputted in a pre-trained model for density-wise classification, and the following results are obtained. The model is successfully able to differentiate different density layers. Figure 20 shows the results obtained by density-wise classification.

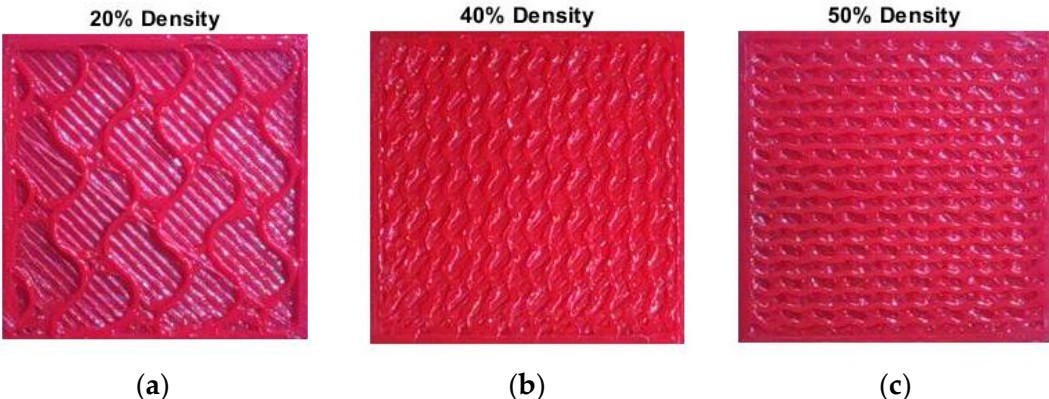

**Figure 20.** Density wise classification: (**a**) Density-wise result showing 20% density for input image; (**b**) Density-wise result showing 40% density for input image; (**c**) Density-wise result showing 50% density for input image.

### 5.4. Error Detection

After a successful comparative study, we implemented the model and algorithm combination for real-time monitoring of components. Results obtained show that Alexnet combined with the SVM algorithm gives maximum accuracy with less computational time, so a combination of Alexnet and SVM is used for real-time monitoring. For error detection, while printing, we captured layer-wise images and fed them into our designed model; the model responded in terms of good and bad for non-defective and defective layers.

The designed pre-trained model algorithm combination separated the defective and non-defective layers as bad and good, respectively. Figure 21 shows the response of the model for the input layer image. Images (d–f) from Figure 21 show a response good for the non-defective layer, and images (a–c) visualize a response for the defective layer, which is bad.

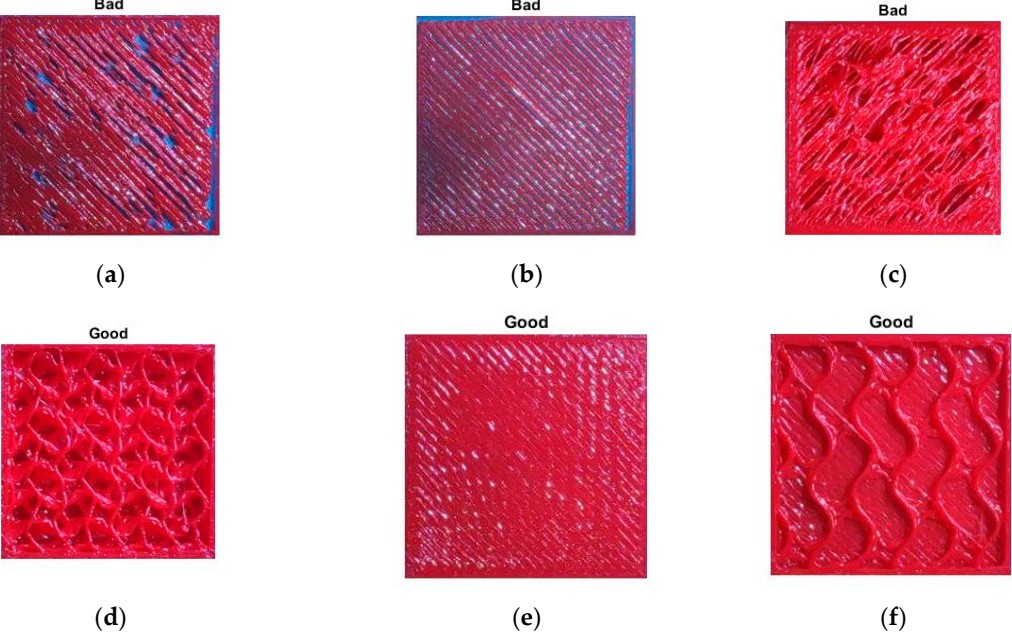

**Figure 21.** Real-time detection: (**a**) Result output for input image as bad layer; (**b**) Result output for input image as bad layer; (**c**) Result output for input image as bad layer; (**d**) Result output for input image as good layer; (**e**) Result output for input image as good layer; (**f**) Result output for input image as good layer.

## 6. Conclusions

The research work carried out in this paper puts forth a comparative analysis of different pre-trained models combined with the ensemble of machine learning algorithms. The study iterates a layer-wise approach for monitoring anomalies in the printed component of a 3D printer. This study aimed to determine which combination gives maximum accuracy in lesser computational time. Density-wise classification using pre-trained models was also carried out. The following conclusions can be derived:

1. A real-time dataset consisting of defective and non-defective 3D printed samples were created for this study. All the AI models were effectively able to perform anomaly classification on this dataset.
2. It was observed from the combination of the pre-trained models with the machine learning techniques that the combination of Alexnet with SVM technique gave the highest accuracy of 99.70%, followed by the combination of Alexnet with K-NN at 99.40%. The other pre-trained models also exhibited decent performance.
3. Further, the analysis of pre-trained models using ensemble learning was carried out to increase the system's accuracy. Alexnet outperformed other pre-trained models by providing 100% accuracy, followed by EfficientNet-B0 at 99.10%.
4. A separate parameter-wise density classification was performed, for which Resnet50 gave an accuracy of 100%.

In the future, the authors propose to implement this fault classification framework on real-time condition monitoring data. This work can be further enhanced by applying explainable fault visualization algorithms to identify the anomalies on images itself. In addition, reinforcement learning can be used improve the model accuracy for finding very fine faults, which are difficult to detect by human eyes.

**Author Contributions:** Conceptualization, V.K. and S.K.; methodology, S.K. and S.W.; software, V.K. and S.K.; validation, A.B., P.K., and S.P.; formal analysis, S.K.; investigation, V.K.; resources, S.K. and A.B.; data curation, S.K.; writing—original draft preparation, V.K., S.K., and A.B.; writing—review and editing, P.K., S.W., S.P., and A.B.; visualization, A.B.; supervision, S.K. and S.W.; project administration, S.K.; funding acquisition, S.K., A.B., P.K., and S.P. All authors have read and agreed to the published version of the manuscript.

**Funding:** This research was funded by Symbiosis Institute of Technology, Symbiosis International (Deemed University) under Research Support Fund (RSF).

**Institutional Review Board Statement:** Not applicable.

**Informed Consent Statement:** Not applicable.

**Data Availability Statement:** The data presented in this study are available on request from the corresponding authors.

**Conflicts of Interest:** The authors declare no conflict of interest.

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
