# Peer review of "Enhancing Surface Fault Detection Using Machine Learning for 3D Printed Products"

_asi, doi:10.3390/asi4020034_

Round 1
Reviewer 1 Report
The paper is within the scope of the article. The presented article deals with using Machine Learning when detecting frauds in 3D printing. The idea of the article is not very modern, but may be interesting for some readers. The topic is also relevant, could have been longer or shorter, but is well understood. A methodology using machine learning algorithms and pre-trained models is also presented.
The introduction is quite well structured, but the authors should think of more defects (Picture 1). Also, the source is missing here... Defect analysis should also be deeper.
A literature review is done only of 9 sources, which is quite a few, based on how popular 3D-printing became...
Methodology is presented in a very good way, conclusions with well explained and proven facts. The paper brings out the new findings of the others.
Materials and methods: Figure 17 not clear. The set-up seems OK to me, measurings also. How can the results be used?
Text quality is medium, text can be understood and is mostly in correct English. The quality of some images is not OK, the tables are all right.
References seem OK.
Author Response
Dear Reviewers,
Thank you for all your valuable suggestions. We have incorporated the same as per your suggestions
|
Sr No. |
Corrections suggested |
Corrections done |
|
Reviewer 1 |
||
|
1 |
The introduction is quite well structured, but the authors should think of more defects (Picture 1). Also, the source is missing here... Defect analysis should also be deeper. |
Thank you for the appreciation. We have removed figure 1 as it was redundant, as all defects are well explained in table. We have carried out more defect analysis is done and reason for neglecting some defect is stated. |
|
2 |
A literature review is done only of 9 sources, which is quite a few, based on how popular 3D-printing became...
|
Yes, keeping in mind this suggestion, more literature is added. Some literature is in descriptive manner and not in tabular form |
|
3 |
Materials and methods: Figure 17 not clear. The set-up seems OK to me, measuring also. How can the results be used?
|
We have removed Fig 17 as it was unnecessary |
Reviewer 2 Report
I think that the research is current. And I agree with the authors, with the sentence they gave in the conclusion, that a detailed qualification of good and bad samples should be presented, as well as the limits when an individual sample is considered good enough.
Figure 19. Sub a) and sub c) refer to the same model of the SVM algorithm? I consider Figure 21 and Figure 23 to be redundant. All required data are clearly seen in Tables 6 and 7.
Author Response
Dear Reviewers,
Thank you for all your valuable suggestions. We have incorporated the same as below:
|
Reviewer 2 |
||
|
1 |
Figure 19. Sub a) and sub c) refer to the same model of the SVM algorithm? I consider Figure 21 and Figure 23 to be redundant. All required data are clearly seen in Tables 6 and 7.
|
Thank you for this suggestion. We have updated with correct labels and removed Fig 21 and 23. |